# THE DEEP WEIGHT PRIOR

**Andrei Atanov**[*]
Skolkovo Institute of Science and Technology
Samsung-HSE Laboratory, National Research
University Higher School of Economics
ai.atanow@gmail.com

**Arsenii Ashukha**[*]
Samsung AI Center Moscow
ars.ashuha@gmail.com

**Kirill Struminsky**
Skolkovo Institute of Science and Technology
National Research University
Higher School of Economics
k.struminsky@gmail.com

**Dmitry Vetrov**
Samsung AI Center Moscow
Samsung-HSE Laboratory, National Research
University Higher School of Economics
vetrovd@yandex.ru

**Max Welling**
University of Amsterdam
Canadian Institute for Advanced Research
m.welling@uva.nl

## ABSTRACT

Bayesian inference is known to provide a general framework for incorporating prior knowledge or specific properties into machine learning models via carefully choosing a prior distribution. In this work, we propose a new type of prior distributions for convolutional neural networks, *deep weight prior (dwp)*, that exploit generative models to encourage a specific structure of trained convolutional filters e.g., spatial correlations. We define *dwp* in a form of an implicit distribution and propose a method for variational inference with such type of implicit priors. In experiments, we show that *dwp* improves the performance of Bayesian neural networks when training data are limited, and initialization of weights with samples from *dwp* accelerates training of conventional convolutional neural networks.

## 1 INTRODUCTION

Bayesian inference is a tool that, after observing training data, allows to transforms a prior distribution over parameters of a machine learning model to a posterior distribution. Recently, stochastic variational inference (Hoffman et al., 2013) – a method for approximate Bayesian inference – has been successfully adopted to obtain a *variational approximation* of a posterior distribution over weights of a deep neural network (Kingma et al., 2015). Currently, there are two major directions for the development of Bayesian deep learning. The first direction can be summarized as the improvement of approximate inference with richer variational approximations and tighter variational bounds (Dikmen et al., 2015). The second direction is the design of probabilistic models, in particular, prior distributions, that widen the scope of applicability of the Bayesian approach.

Prior distributions play an important role for sparsification (Molchanov et al., 2017; Neklyudov et al., 2017), quantization (Ullrich et al., 2017) and compression (Louizos et al., 2017; Federici et al., 2017) of deep learning models. Although these prior distributions proved to be helpful, they are limited to fully-factorized structure. Thus, the often observed spatial structure of convolutional filters cannot be enforced with such priors. Convolutional neural networks are an example of the model family, where a correlation of the weights plays an important role, thus it may benefit from more flexible prior distributions.

Convolutional neural networks are known to learn similar convolutional kernels on different datasets from similar domains (Sharif Razavian et al., 2014; Yosinski et al., 2014). Based on this fact, within

---

[*]Equal contribution

a specific data domain, we consider a distribution of convolution kernels of trained convolutional networks. In the rest of the paper, we refer to this distribution as the *source kernel distribution*. Our main assumption is that within a specific domain the source kernel distribution can be efficiently approximated with convolutional kernels of models that were trained on a small subset of problems from this domain. For example, given a specific architecture, we expect that kernels of a model trained on notMNIST dataset – a dataset of grayscale images – come from the same distribution as kernels of the model trained on MNIST dataset. In this work, we propose a method that estimates the source kernel distribution in an implicit form and allows us to perform variational inference with the specific type of implicit priors.

Our contributions can be summarized as follows:

1. We propose *deep weight prior*, a framework that approximates the *source kernel distribution* and incorporates prior knowledge about the structure of convolutional filters into the prior distribution. We also propose to use an implicit form of this prior (Section 3.1).

2. We develop a method for variational inference with the proposed type of implicit priors (Section 3.2).

3. In experiments (Section 4), we show that variational inference with *deep weight prior* significantly improves classification performance upon a number of popular prior distributions in the case of limited training data. We also find that initialization of conventional convolution networks with samples from a deep weight prior leads to faster convergence and better feature extraction without training i.e., using random weights.

## 2   DEEP BAYES

In Bayesian setting, after observing a dataset $\mathcal{D} = \{x_1, \ldots, x_N\}$ of $N$ points, the goal is to transform our prior knowledge $p(\omega)$ of the unobserved distribution parameters $\omega$ to the posterior distribution $p(\omega \,|\, \mathcal{D})$. However, computing the posterior distribution through Bayes rule $p(\omega \,|\, \mathcal{D}) = p(\mathcal{D} \,|\, \omega)p(\omega)/p(\mathcal{D})$ may involve computationally intractable integrals. This problem, nonetheless, can be solved approximately.

Variational Inference (Jordan et al., 1999) is one of such approximation methods. It reduces the inference to an optimization problem, where we optimize parameters $\theta$ of a variational approximation $q_\theta(\omega)$, so that KL-divergence between $q_\theta(\omega)$ and $p(\omega \,|\, \mathcal{D})$ is minimized. This divergence in practice is minimized by maximizing the *variational lower bound* $\mathcal{L}(\theta)$ of the marginal log-likelihood of the data w.r.t parameters $\theta$ of the variational approximation $q_\theta(W)$.

$$\mathcal{L}(\theta) = L_\mathcal{D} - D_{\mathrm{KL}}(q_\theta(\omega)\|p(\omega)) \to \max_\theta \tag{1}$$

$$\text{where} \quad L_\mathcal{D} = \mathbb{E}_{q_\theta(\omega)} \log p(D \,|\, \omega) \tag{2}$$

The variational lower bound $\mathcal{L}(\theta)$ consists of two terms: 1) the (conditional) *expected log likelihood* $L_\mathcal{D}$, and 2) the regularizer $D_{\mathrm{KL}}(q_\theta(\omega)\|p(\omega))$. Since $\log p(\mathcal{D}) = \mathcal{L}(\theta) + D_{\mathrm{KL}}(q_\theta(\omega)\|p(\omega \,|\, \mathcal{D}))$ and $p(\mathcal{D})$ does not depend on $q_\theta(w)$ maximizing of $\mathcal{L}(\theta)$ minimizes $D_{\mathrm{KL}}(q_\theta(\omega)\|p(\omega \,|\, \mathcal{D}))$. However, in case of intractable expectations in equation 1 neither the variational lower bound $\mathcal{L}(\theta)$ nor its gradients can be computed in a closed form.

Recently, Kingma & Welling (2013) and Rezende et al. (2014) proposed an efficient mini-batch based approach to stochastic variational inference, so-called *stochastic gradient variational Bayes* or *doubly stochastic variational inference*. The idea behind this framework is reparamtetrization, that represents samples from a parametric distribution $q_\theta(\omega)$ as a deterministic differentiable function $\omega = f(\theta, \epsilon)$ of parameters $\theta$ and an (auxiliary) noise variable $\epsilon \sim p(\epsilon)$. Using this trick we can efficiently compute an unbiased stochastic gradient $\nabla_\theta \mathcal{L}$ of the variational lower bound w.r.t the parameters of the variational approximation.

**Bayesian Neural Networks.** The stochastic gradient variational Bayes framework has been applied to approximate posterior distributions over parameters of deep neural networks (Kingma et al., 2015). We consider a discriminative problem, where dataset $\mathcal{D}$ consists of $N$ object-label pairs $\mathcal{D} = \{(x_i, y_i)\}_{i=1}^N$. For this problem we maximize the variational lower bound $\mathcal{L}(\theta)$ with respect to

parameters $\theta$ of a variational approximation $q_\theta(W)$:

$$\mathcal{L}(\theta) = \sum_{i=1}^{N} \mathbb{E}_{q_\theta(W)} \log p(y_i \,|\, x_i, W) - D_{\mathrm{KL}}(q_\theta(W) \| p(W)) \to \max_\theta \qquad (3)$$

where $W$ denotes weights of a neural network, $q_\theta(W)$ is a variational distribution, that allows reparametrization (Kingma & Welling, 2013; Figurnov et al., 2018) and $p(W)$ is a prior distribution. In the simplest case $q_\theta(W)$ can be a fully-factorized normal distribution. However, more expressive variational approximations may lead to better quality of variational inference (Louizos & Welling, 2017; Yin & Zhou, 2018). Typically, Bayesian neural networks use fully-factorized normal or log-uniform priors (Kingma et al., 2015; Molchanov et al., 2017; Louizos & Welling, 2017).

**Variational Auto-encoder.** Stochastic gradient variational Bayes has also been applied for building generative models. The variational auto-encoder proposed by Kingma & Welling (2013) maximizes a variational lower bound $\mathcal{L}(\theta, \phi)$ on the marginal log-likelihood by amortized variational inference:

$$\mathcal{L}(\theta, \phi) = \sum_{i=1}^{N} \mathbb{E}_{q_\theta(z_i \,|\, x_i)} \log p_\phi(x_i \,|\, z_i) - D_{\mathrm{KL}}(q_\theta(z_i \,|\, x_i) \| p(z_i)) \to \max_{\theta, \phi}, \qquad (4)$$

where an *inference model* $q_\theta(z_i \,|\, x_i)$ approximates the posterior distribution over local latent variables $z_i$, *reconstruction model* $p_\phi(x_i \,|\, z_i)$ transforms the distribution over latent variables to a conditional distribution in object space and a prior distribution over latent variables $p(z_i)$. The vanilla VAE defines $q_\theta(z \,|\, x)$, $p_\phi(x \,|\, z)$, $p(z)$ as fully-factorized distributions, however, a number of richer variational approximations and prior distributions have been proposed (Rezende & Mohamed, 2015; Kingma et al., 2016; Tomczak & Welling, 2017). The approximation of the data distribution can then be defined as an intractable integral $p(x) \approx \int p_\phi(x \,|\, z) p(z) \, dz$ which we will refer to as an implicit distribution.

## 3   DEEP WEIGHT PRIOR

In this section, we introduce the *deep weight prior* – an expressive prior distribution that is based on generative models. This prior distribution allows us to encode and favor the structure of learned convolutional filters. We consider a neural network with $L$ convolutional layers and denote parameters of $l$-th convolutional layer as $w^l \in \mathbb{R}^{I_l \times O_l \times H_l \times W_l}$, where $I_l$ is the number of input channels, $O_l$ is the number of output channels, $H_l$ and $W_l$ are spatial dimensions of kernels. Parameters of the neural network are denoted as $W = (w^1, \dots w^L)$. A variational approximation $q_\theta(W)$ and a prior distribution $p(W)$ have the following factorization over layers, filters and channels:

$$q_\theta(W) = \prod_{l=1}^{L} \prod_{i=1}^{I_l} \prod_{j=1}^{O_l} q(w_{ij}^l \,|\, \theta_{ij}^l) \qquad p(W) = \prod_{l=1}^{L} \prod_{i=1}^{I_l} \prod_{j=1}^{O_l} p_l(w_{ij}^l), \qquad (5)$$

where $w_{ij}^l \in \mathbb{R}^{H_l \times W_l}$ is a kernel of $j$-th channel in $i$-th filter of $l$-th convolutional layer. We also assume that $q_\theta(W)$ allows reparametrization. The prior distribution $p(W)$, in contrast to popular prior distributions, is not factorized over spatial dimensions of the filters $H_l, W_l$.

For a specific data domain and architecture, we define the *source kernel distribution* – the distribution of trained convolutional kernels of the $l$-th convolutional layer. The source kernel distribution favors learned kernels, and thus it is a very natural candidate to be the prior distribution $p_l(w_{ij}^l)$ for convolutional kernels of the $l$-th layer. Unfortunately, we do not have access to its probability density function (p.d.f.), that is needed for most approximate inference methods e.g., variational inference. Therefore, we assume that the p.d.f. of the source kernel distribution can be approximated using kernels of models trained on external datasets from the same domain. For example, given a specific architecture, we expect that kernels of a model trained on CIFAR-100 dataset come from the same distribution as kernels of the model trained on CIFAR-10 dataset. In other words, the p.d.f. of the source kernel distribution can be approximated using a small subset of problems from a specific data domain. In the next subsection, we propose to approximate this intractable probability density function of the source kernel distribution using the framework of generative models.

---

**Algorithm 1** Stochastic Variational Inference With Implicit Prior Distribution

---

**Require:** dataset $\mathcal{D} = \{(x_i, y_i)\}_{i=1}^N$
**Require:** variational approximations $q(w \mid \theta_{ij}^l)$ and reverse models $r(z \mid w; \psi_l)$
**Require:** reconstruction models $p(w \mid z; \phi_l)$, priors for auxiliary variables $p_l(z)$
 **while** not converged **do**
  $\hat{M} \leftarrow$ mini-batch of objects form dataset $\mathcal{D}$
  $\hat{w}_{ij}^l \leftarrow$ sample weights from $q(w|\theta_{ij}^l)$ with reparametrization
  $\hat{z}_{ij}^l \leftarrow$ sample auxiliary variables from $r(z \mid \hat{w}_{ij}^l; \psi_l)$ with reparametrization
  $\hat{\mathcal{L}}^{aux} \leftarrow L_{\hat{M}} + \sum_{l,i,j} -\log q(\hat{w}_{ij}^l \mid \theta_{ij}^l) - \log r(\hat{z}_{ij}^l \mid \hat{w}_{ij}^l; \psi_l) + \log p_l(\hat{z}_{ij}^l) + \log p(\hat{w}_{ij}^l \mid \hat{z}_{ij}^l; \phi_l)$
  Obtain unbiased estimate $\hat{g}$ with $\mathbb{E}[\hat{g}] = \nabla \mathcal{L}^{aux}$ by differentiating $\hat{\mathcal{L}}^{aux}$
  Update parameters $\theta$ and $\psi$ using gradient $\hat{g}$ and a stochastic optimization algorithm
 **end while**
 **return** Parameters $\theta$, $\psi$

---

## 3.1 Model of Prior Distribution

In this section, we discuss explicit and implicit approximations $\hat{p}_l(w)$ of the probability density function $p_l(w)$ of the source kernel distribution of $l$-th layer. We assume to have a trained convolutional neural network, and treat kernels from the $l$-th layer of this network $w_{ij}^l \in R^{H_l \times W_l}$ as samples from the source kernel distribution of $l$-th layer $p_l(w)$.

**Explicit models.** A number of approximations allow us to evaluate probability density functions explicitly. Such families include but are not limited to Kernel Density Estimation (Silverman, 1986), Normalizing Flows (Rezende & Mohamed, 2015; Dinh et al., 2017) and PixelCNN (van den Oord et al., 2016). For these families, we can estimate the KL-divergence $D_{KL}(q(w \mid \theta_{ij}^l) \| \hat{p}_l(w))$ and its gradients without a systematic bias, and then use them for variational inference. Despite the fact that these methods provide flexible approximations, they usually demand high memory or computational cost (Louizos & Welling, 2017).

**Implicit models.** Implicit models, in contrast, can be more computationally efficient, however, they do not provide access to an explicit form of probability density function $\hat{p}_l(w)$. We consider an approximation of the prior distribution $p_l(w)$ in the following implicit form:

$$\hat{p}_l(w) = \int p(w \mid z; \phi_l) p_l(z) \, dz, \tag{6}$$

where a conditional distribution $p(w \mid z; \phi_l)$ is an explicit parametric distribution and $p_l(z)$ is an explicit prior distribution that does not depend on trainable parameters. Parameters of the conditional distribution $p(w \mid z; \phi_l)$ can be modeled by a differentiable function $g(z; \phi_l)$ e.g. neural network. Note, that while the conditional distribution $p(w \mid z; \phi_l)$ usually is a simple explicit distribution, e.g. fully-factorized Gaussian, the marginal distribution $\hat{p}_l(w)$ is generally a more complex intractable distribution.

Parameters $\phi_l$ of the conditional distribution $p(w \mid z; \phi_l)$ can be fitted using the variational auto-encoder framework. In contrast to the methods with explicit access to the probability density, variational auto-encoders combine low memory cost and fast sampling. However, we cannot obtain an unbiased estimate the logarithm of probability density function $\log \hat{p}_l(w)$ and therefore cannot build an unbiased estimator of the variational lower bound (equation 3). In order to overcome this limitation we propose a modification of variational inference for implicit prior distributions.

## 3.2 Variational Inference With Implicit Prior Distribution

Stochastic variational inference approximates a true posterior distribution by maximizing the variational lower bound $\mathcal{L}(\theta)$ (equation 1), which includes the KL-divergence $D_{KL}(q(W) \| p(W))$ between a variational approximation $q_\theta(W)$ and a prior distribution $p(W)$. In the case of simple prior and variational distributions (e.g. Gaussian), the KL-divergence can be computed in a closed form or unbiasedly estimated. Unfortunately, it does not hold anymore in case of an implicit prior distribution $\hat{p}(W) = \Pi_{l,i,j} \hat{p}_l(w_{ij}^l)$. In that case, the KL-divergence cannot be estimated without bias.

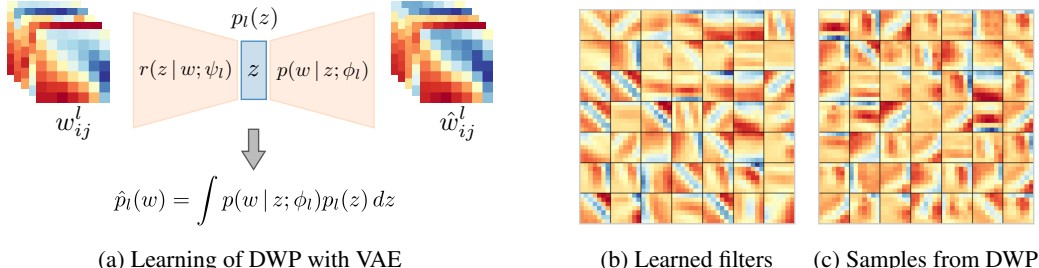

(a) Learning of DWP with VAE    (b) Learned filters    (c) Samples from DWP

Figure 1: At subfig. 1(a) we show the process of learning the prior distribution over kernels of one convolutional layer. First, we train encoder $r(z \mid w; \phi_l)$ and decoder $p(w \mid z; \psi_l)$ with VAE framework. Then, we use the decoder to construct the prior $\hat{p}_l(w)$. At subfig. 1(b) we show a batch of learned kernels of shape $7 \times 7$ form the first convolutional layer of a CNN trained on NotMNIST dataset, at subfig. 1(c) we show samples form the deep weight prior that is learned on these kernels.

To make the computation of the variational lower bound tractable, we introduce an auxiliary lower bound on the KL-divergence. KL-divergence:

$$D_{\mathrm{KL}}(q(W)\|\hat{p}(W)) = \sum_{l,i,j} D_{\mathrm{KL}}(q(w_{ij}^l|\theta_{ij}^l)\|\hat{p}_l(w_{ij}^l)) \leq \sum_{l,i,j} \big( - H(q(w_{ij}^l \mid \theta_{ij}^l)) +$$

$$+ \mathbb{E}_{q(w_{ij}^l \mid \theta_{ij}^l)} \big[ D_{\mathrm{KL}}(r(z \mid w_{ij}^l; \psi_l)\|p_l(z)) - \mathbb{E}_{r(z \mid w_{ij}^l; \psi_l)} \log p(w_{ij}^l \mid z; \phi_l) \big] \big) = D_{\mathrm{KL}}^{bound}, \quad (7)$$

where $r(z \mid w; \psi_l)$ is an auxiliary inference model for the prior of $l$-th layer $\hat{p}_l(w)$, The final auxiliary variational lower bound has the following form:

$$\mathcal{L}^{aux}(\theta, \psi) = L_D - D_{\mathrm{KL}}^{bound} \leq L_D - D_{\mathrm{KL}}(q_\theta(W)\|\hat{p}(W)) = \mathcal{L}(\theta) \quad (8)$$

The lower bound $\mathcal{L}^{aux}$ is tight if and only if the KL-divergence between the auxiliary reverse model and the intractable posterior distribution over latent variables $z$ given $w$ is zero (Appendix A).

In the case when $q_\theta(w)$, $p(w \mid z; \phi_l)$ and $r(z \mid w; \psi_l)$ are explicit parametric distributions which can be reparametrized, we can perform an unbiased estimation of a gradient of the auxiliary variational lower bound $\mathcal{L}^{aux}(\theta, \psi)$ (equation 8) w.r.t. parameters $\theta$ of the variational approximation $q_\theta(W)$ and parameters $\psi$ of the reverse models $r(z \mid w; \psi_l)$. Then we can maximize the auxiliary lower bound w.r.t. parameters of the variational approximation and the reversed models $\mathcal{L}^{aux}(\theta, \psi) \to \max_{\theta, \psi}$. Note, that parameters $\phi$ of the prior distribution $\hat{p}(W)$ are fixed during variational inference, in contrast to the Empirical Bayesian framework (MacKay, 1992).

Algorithm 1 describes stochastic variational inference with an implicit prior distribution. In the case when we can calculate an entropy $H(q)$ or the divergence $D_{\mathrm{KL}}(r(z \mid w; \psi_l)\|p_l(z))$ explicitly, the variance of the estimation of the gradient $\nabla \hat{\mathcal{L}}^{aux}(\theta, \psi)$ can be reduced. This algorithm can also be applied to an implicit prior that is defined in the form of Markov chain:

$$\hat{p}(w) = \int dz_0 \ldots dz_T p(w \mid z_T) p(z_0) \prod_{t=0}^{T-1} p(z_{t+1} \mid z_t), \quad (9)$$

where $p(z_{t+1} \mid z_t)$ is a transition operator (Salimans et al., 2015), see Appendix A. We provide more details related to the form of $p(w \mid z; \phi_l)$, $r(z \mid w; \psi_l)$ and $p_l(z)$ distributions in Section 4.

### 3.3    LEARNING DEEP WEIGHT PRIOR

In this subsection we explain how to train deep weight prior models for a particular problem. We present samples from learned prior distribution at Figure 1(c).

**Source datasets of kernels.** For kernels of a particular convolutional layer $l$, we train an individual prior distribution $\hat{p}_l(w) = \int p(w \mid z; \phi_l) p_l(z) \, dz$. First, we collect a source dataset of the kernels of the $l$-th layer of convolutional networks (source networks) trained on a dataset from a similar domain. Then, we train reconstruction models $p(w \mid z; \phi_l)$ on these collected source datasets for

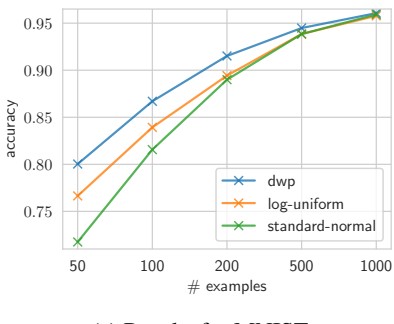

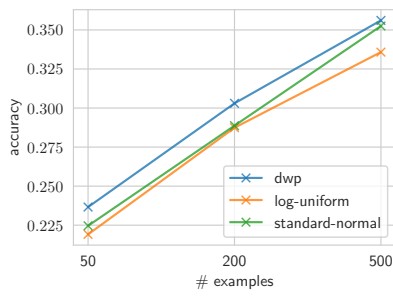

(a) Results for MNIST

(b) Results for CIFAR-10

Figure 2: For different sizes of training set of MNIST and CIFAR-10 datasets, we demonstrate the performance of variational inference with a fully-factorized variational approximation with three different prior distributions: deep weight prior (dwp), log-uniform, and standard normal. We found that variational inference with a deep weight prior distribution achieves better mean test accuracy comparing to learning with standard normal and log-uniform prior distributions.

each layer, using the framework of variational auto-encoder (Section 2). Finally, we use the reconstruction models to construct priors $\hat{p}_l(w)$ as shown at Figure 1(a). In our experiments, we found that regularization is crucial for learning of source kernels. It helps to learn more structured and less noisy kernels. Thus, source models were learned with $L_2$ regularization. We removed kernels of small norm as they have no influence upon predictions (Molchanov et al., 2017), but they make learning of the generative model more challenging.

**Reconstruction and inference models for prior distribution.** In our experiments, inference models $r(z \mid w; \psi_l)$ are fully-factorized normal-distributions $\mathcal{N}(z \mid \mu_{\psi_l}(w), \mathrm{diag}(\sigma^2_{\psi_l}(w)))$, where parameters $\mu_{\psi_l}(w)$ and $\sigma_{\psi_l}(w)$ are modeled by a convolutional neural network. The convolutional part of the network is constructed from several convolutional layers that are alternated with ELU (Clevert et al., 2015) and max-pooling layers. Convolution layers are followed by a fully-connected layer with $2 \cdot z_{dim}^l$ output neurons, where $z_{dim}^l$ is a dimension of the latent representation $z$, and is specific for a particular layer.

Reconstruction models $p(w \mid z; \phi_l)$ are also modeled by a fully-factorized normal-distribution $\mathcal{N}(w \mid \mu_{\phi_l}(z), \mathrm{diag}(\sigma^2_{\phi_l}(z)))$ and network for $\mu_{\phi_l}$ and $\sigma^2_{\phi_l}$ has the similar architecture as the inference model, but uses transposed convolutions. We use the same architectures for all prior models, but with slightly different hyperparameters, due to different sizes of kernels. We also use fully-factorized standard Gaussian prior $p_l(z_i) = \mathcal{N}(z_i \mid 0, 1)$ for latent variables $z_i$. We provide a more detailed description at Appendix F.

## 4 EXPERIMENTS

We apply deep weight prior to variational inference, random feature extraction and initialization of convolutional neural networks. In our experiments we used MNIST (LeCun et al., 1998), NotMNIST (Bulatov, 2011), CIFAR-10 and CIFAR-100 (Krizhevsky & Hinton, 2009) datasets. Experiments were implemented[1] using PyTorch (Paszke et al., 2017). For optimization we used Adam (Kingma & Ba, 2014) with default hyperparameters. We trained prior distributions on a number of source networks which were learned from different initial points on NotMNIST and CIFAR-100 datasets for MNIST and CIFAR-10 experiments respectively.

### 4.1 CLASSIFICATION

In this experiment, we performed variational inference over weights of a discriminative convolutional neural network (Section 3) with three different prior distributions for the weights of the convolutional layers: deep weight prior (dwp), standard normal and log-uniform (Kingma et al., 2015). We did not perform variational inference over the parameters of the fully connected layers. We used

---

[1] The code is available at `https://github.com/bayesgroup/deep-weight-prior`

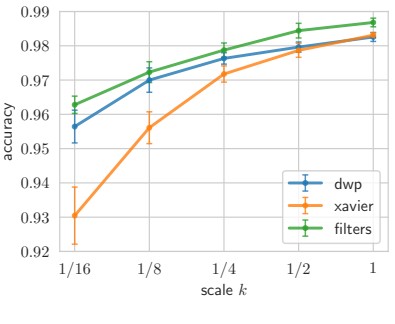 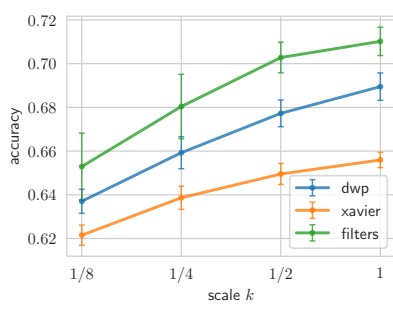

        (a) Results for MNIST                      (b) Results for CIFAR-10

Figure 3: We study the influence of initialization of convolutional filters on the performance of random feature extraction. In the experiment, the weights of convolutional filters were initialized randomly and fixed. The initializations were sampled from deep weight prior (dwp), learned filters (filters) and samples from Xavier distribution (xavier). We performed the experiment for different size of the model, namely, to obtain models of different sizes we scaled a number of filters in all convolutional layers linearly by $k$. For every size of the model, we averaged results by 10 runs. We found that initialization with samples from deep weight prior and learned filters significantly outperform Xavier initialization. Although, initialization with filters performs marginally better, *dwp* does not require to store a potentially big set of all learned filters. We present result for MNIST and CIFAR-10 datasets at sub figs. 3(a) and 3(b) respectively.

a fully-factorized variational approximation with additive parameterization proposed by Molchanov et al. (2017) and local reparametrization trick proposed by Kingma et al. (2015). Note, that our method can be combined with more complex variational approximations, in order to improve variational inference.

On MNIST dataset we used a neural network with two convolutional layers with $32, 128$ filters of shape $7 \times 7$, $5 \times 5$ respectively, followed by one linear layer with $10$ neurons. On the CIFAR dataset we used a neural network with four convolutional layers with $128$, $256$, $256$ filters of shape $7 \times 7$, $5 \times 5$, $5 \times 5$ respectively, followed by two fully connected layers with $512$ and $10$ neurons. We used a max-pooling layer (Nagi et al., 2011) After the first convolutional layer. All layers were divided with leaky ReLU nonlinearities (Nair & Hinton, 2010).

At figure 2 we report accuracy for variational inference with different sizes of training datasets and prior distributions. Variational inference with deep weight prior leads to better mean test accuracy, in comparison to log-uniform and standard normal prior distributions. Note that the difference gets more significant as the training set gets smaller.

## 4.2 Random Feature Extraction

Convolutional neural networks produce useful features even if they are initialized randomly (Saxe et al., 2011; He et al., 2016; Ulyanov et al., 2017). In this experiment, we study an influence of different random initializations of convolutional layers – that is fixed during training – on the performance of convolutional networks of different size, where we train only fully-connected layers. We use three initializations for weights of convolutional layers: learned kernels, samples from deep weight prior, samples from Xavier distribution (Glorot & Bengio, 2010). We use the same architectures as in Section 4.1. We found that initializations with samples from deep weight prior and learned kernels significantly outperform the standard Xavier initialization when the size of the network is small. Initializations with samples form deep weight prior and learned filters perform similarly, but with deep weight prior we can avoid storing all learned kernels. At Figure 3, we show results on MNIST and CIFAR-10 for different network sizes, which are obtained by scaling the number of filters by $k$.

## 4.3 Convergence

Deep learning models are sensitive to initialization of model weights. In particular, it may influence the speed of convergence or even a local minimum a model converges to. In this experiment, we study the influence of initialization on the convergence speed of two settings: a variational auto-

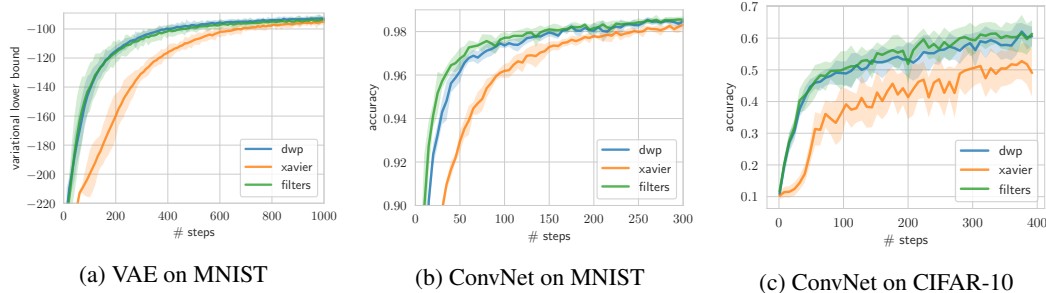

|   |   |   |
|---|---|---|
| (a) VAE on MNIST | (b) ConvNet on MNIST | (c) ConvNet on CIFAR-10 |

Figure 4: We found that initialization of weights of the models with deep weight priors or learned filters significantly increases the training speed, comparing to Xavier initialization. At subplot 4(a) we report a variational lower bound for variational auto-encoder, at subplots 4(b) and 4(c) we report accuracy for convolution networks on MINTS and CIFAR-10.

encoder on MNIST, and convolutional networks on MNIST and CIFAR-10. We compare three different initializations of weights of conventional convolutional layers: learned filters, samples from deep weight prior and samples form Xavier distribution.

Figure 4 provides the results for a convolutional variational auto-encoder trained on MNIST and for a convolutional classification network trained on CIFAR-10 and MNIST. We found that deep weight prior and learned filters initializations perform similarly and lead to significantly faster convergence comparing to standard Xavier initialization. Deep weight prior initialization however does not require us to store a possibly large set of filters. Also, we plot samples from variational auto-encoders at a different training steps Appendix E.

## 5  RELATED WORK

The recent success of transfer learning (Yosinski et al., 2014) shows that convolutional networks produce similar convolutional filters while being trained on different datasets from the same domain e.g. photo-realistic images. In contrast to Bayesian techniques (Kingma et al., 2015; Kochurov et al., 2018), these methods do not allow to obtain a posterior distribution over parameters of the model, and in most cases, they require to store convolutional weights of pre-trained models and careful tuning of hyperparameters.

The Bayesian approach provides a framework that incorporates prior knowledge about weights of a machine learning model by choosing or leaning a prior distribution $p(w)$. There is a huge amount of works on prior distributions for Bayesian inference (MacKay, 1992; Williams, 1995), where empirical Bayes – an approach that tunes parameters of the prior distribution on the training data – plays an important role (MacKay, 1992). These methods are widely used for regularization and sparsification of linear models (Bishop & Tipping, 2003), however, applied to deep neural networks (Kingma et al., 2015; Ullrich et al., 2017), they do not take into account the structure of the model weights, e.g. spatial correlations, which does matter in case of convolutional networks. Our approach allows to perform variational inference with an implicit prior distribution, that is based on previously observed convolutional kernels. In contrast to an empirical Bayes approach, parameters $\phi$ of a deep weight prior (equation 6) are adjusted before the variational inference and then remain fixed.

Prior to our work implicit models have been applied to variational inference. That type of models includes a number of flexible variational distributions e.g., semi-implicit (Yin & Zhou, 2018) and Markov chain (Salimans et al., 2015; Lamb et al., 2017) approximations. Implicit priors have been used for introducing invariance properties (Nalisnick & Smyth, 2018), improving uncertainty estimation (Ma et al., 2018) and learning meta-representations within an empirical Bayes approach (Karaletsos et al., 2018).

In this work, we propose to use an implicit prior distribution for stochastic variational inference (Kingma et al., 2015) and develop a method for variational inference with the specific type of implicit priors. The approach also can be generalized to prior distributions in the form of a Markov chain. We show how to use this framework to learn a flexible prior distribution over kernels of Bayesian convolutional neural networks.

## 6 DISCUSSION & CONCLUSION

In this work we propose *deep weight prior* – a framework for designing a prior distribution for convolutional neural networks, that exploits prior knowledge about the structure of learned convolutional filters. This framework opens a new direction for applications of Bayesian deep learning, in particular to transfer learning.

**Factorization.** The factorization of deep weight prior does not take into account inter-layer dependencies of the weights. Although a more complex factorization might be a better fit for CNNs. Accounting inter-layer dependencies may give us an opportunity to recover a distribution in the space of trained networks rather than in the space of trained kernels. However, estimating prior distributions of more complex factorization may require significantly more data and computational budget, thus the topic needs an additional investigation.

**Inference.** An alternative to variational inference with auxiliary variables (Salimans et al., 2015) is semi-implicit variational inference (Yin & Zhou, 2018). The method was developed only for semi-implicit variational approximations, and only the recent work on doubly semi-implicit variational inference generalized it for implicit prior distributions (Molchanov et al., 2018). These algorithms might provide a better way for variational inference with a deep weight prior, however, the topic needs further investigation.

ACKNOWLEDGMENTS

We would like to thank Ekaterina Lobacheva, Dmitry Molchanov, Kirill Neklyudov and Ivan Sosnovik for valuable discussions and feedback on the earliest version of this paper. Andrei Atanov was supported by Samsung Research, Samsung Electronics. Kirill Struminsky was supported by Ministry of Education and Science of the Russian Federation (grant 14.756.31.0001).

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

## A  Variational Inference with Implicit Prior Distribution

We consider a variational lower bound $\mathcal{L}$ with variational approximation $q(w)$ and prior distribution defined in a form of Markov chain $p(w) = \int dz_0 \ldots dz_T p(w \mid z_T) \prod_{t=0}^T p(z_{t+1} \mid z_t) p(z_0)$ and joint distribution $p(w, \boldsymbol{z}) = p(w \mid z_T) \prod_{t=0}^T p(z_{t+1} \mid z_t) p(z_0)$. Where $p(z_{t+1} \mid z_t)$ is a transition operator, and $\boldsymbol{z} = (z_0, \ldots, z_T)$ (Salimans et al., 2015). Unfortunately, gradients of $\mathcal{L}$ cannot be efficiently estimated, but we construct a tractable lower bound $\mathcal{L}^{aux}$ for $\mathcal{L}$:

$$\mathcal{L} = \mathbb{E}_{q(w)} \left[ \log p(x \mid w) p(w) - \log q(w) \right] = \mathbb{E}_{q(w)} \mathbb{E}_{r(\boldsymbol{z} \mid w)} \left[ \log p(x \mid w) p(w) - \log q(w) \right] = \tag{10}$$

$$= \mathbb{E}_{q(w)} \mathbb{E}_{r(\boldsymbol{z} \mid w)} \left[ \log p(x \mid w) \frac{p(w, \boldsymbol{z})}{p(\boldsymbol{z} \mid w)} \frac{r(\boldsymbol{z} \mid w)}{r(\boldsymbol{z} \mid w)} - \log q(w) \right] = \tag{11}$$

$$= \mathbb{E}_{q(w)} \mathbb{E}_{r(\boldsymbol{z} \mid w)} \left[ \log p(x \mid w) \frac{p(w, \boldsymbol{z})}{r(\boldsymbol{z} \mid w)} - \log q(w) \right] + \mathbb{E}_{q(w)} D_{\mathrm{KL}}(r(\boldsymbol{z} \mid w) \| p(\boldsymbol{z} \mid w)) = \tag{12}$$

$$= \mathcal{L}^{aux} + \mathbb{E}_{q(w)} D_{\mathrm{KL}}(r(\boldsymbol{z} \mid w) \| p(\boldsymbol{z} \mid w)) \geq \mathcal{L}^{aux}. \tag{13}$$

Inequality 13 has a very natural interpretation. The lower bound $\mathcal{L}^{aux}$ is tight if and only if the KL-divergence between the auxiliary reverse model and the posterior intractable distribution $p(z \mid w)$ is zero.

The deep weight prior (Section 3) is a special of Markov chain prior for $T = 0$ and $p(w) = \int p(w \mid z) p(z) dz$. The auxiliary variational bound has the following form:

$$\mathcal{L}^{aux} = \mathbb{E}_{q(w)} \mathbb{E}_{r(z \mid w)} \left[ \log p(x \mid w) \frac{p(w \mid z) p(z)}{r(z \mid w)} - \log q(w) \right] = \tag{14}$$

$$= \mathbb{E}_{q(w)} \left[ \log p(x \mid w) \right] + H(q) - \mathbb{E}_{q(w)} \left[ D_{\mathrm{KL}}(r(z \mid w) \| p(z) - \mathbb{E}_{r(z \mid w)} \log p(w \mid z)) \right]. \tag{15}$$

where the gradients in equation 14 can be efficiently estimated in case $q(w)$, for explicit distributions $q(w), p_\phi(w \mid z), r(z \mid w)$ that can be reparametrized.

## B  The Estimate of the Approximation Gap with IWAE Estimates

| nats, $\times 10^3$ | $\mathcal{L}^{aux}(\theta, \psi)$ | $\mathcal{L}_{10000}^{IWAE}(\theta, \psi)$ | $G(\theta, \psi) \geq$ | $\mathcal{L}^{aux}(\theta, \psi_{p_l})$ |
|---|---|---|---|---|
| | $-23.375 \pm 0.230$ | $-9.957$ | $13.418$ | $-128.325 \pm 2.436$ |

Table 1: Comparison of the proposed auxiliary lower bound with IWAE lower bound estimation.

During variational inference with deep weight prior (Algorithm 1) we optimize a new auxiliary lower bound $\mathcal{L}^{aux}(\theta, \psi)$ on the evidence lower bound $\mathcal{L}(\theta)$. However, the quality of such inference depends on the gap $G(\theta, \psi)$ between the original variational lower bound $\mathcal{L}(\theta)$ and the variational lower bound in auxiliary space $\mathcal{L}^{aux}(\theta, \psi)$:

$$G(\theta, \psi) = \mathcal{L}(\theta) - \mathcal{L}^{aux}(\theta, \psi). \tag{16}$$

The gap $G(\theta, \psi)$ cannot be calculated exactly, but it can be estimated by using tighter but less computationally efficient lower bound. We follow Burda et al. (2015) and construct tighter lower bound $\mathcal{L}_K^{IWAE}(\theta, \psi)$:

$$\mathcal{L}_K^{IWAE}(\theta, \psi) = L_D + H(q(W)) + \tag{17}$$

$$+ \sum_{l,i,j} \mathbb{E}_{q(w_{ij}^l \mid \theta_{ij}^l)} \mathbb{E}_{z_1, \ldots, z_K \sim q(z \mid w_{ij}^l \psi_l)} \log \left( \frac{1}{K} \sum_{k=1}^K \frac{p(w_{ij}^l \mid z_k) p_l(z_k)}{q(z_k \mid w_{ij}^l \psi_l)} \right). \tag{18}$$

The estimate $\mathcal{L}_K^{IWAE}(\theta, \psi)$ converges to $\mathcal{L}(\theta)$ with $K$ goes to infinity (Burda et al., 2015). We estimate the gap with $K = 10000$ as follows:

$$G(\theta, \psi) \geq \mathcal{L}_{10000}^{IWAE}(\theta, \psi) - \mathcal{L}^{aux}(\theta, \psi). \tag{19}$$

The results are presented at the Table 1. In order to show the range of the estimate and the gain from learning of $q(z \mid w_{ij}^l \psi_l)$ we compare results to the value of auxiliary lower bound $\mathcal{L}^{aux}(\theta, \psi_{p_l})$ computed at the point $\psi_{p_l}$ where $q(z \mid w_{ij}^l \psi_{p_l}) \equiv p_l(z)$. The estimate of the gap, however, may be not very accurate and we consider it as a sanity check.

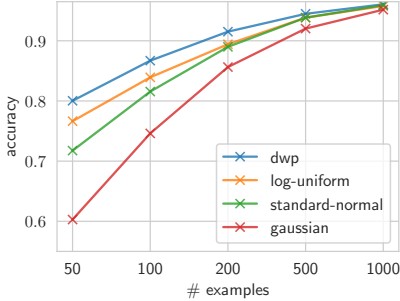

Figure 5: For different sizes of training set of MNIST dataset, we demonstrate the performance of variational inference with a fully-factorized variational approximation with three different prior distributions: deep weight prior (dwp), log-uniform, standard normal and learned multivariate gaussian. For more details see Section 4.1.

## C  UNIVARIATE GAUSSIAN PRIOR

We examined a multivariate normal distribution $\hat{p}_l(w) = \mathcal{N}(w|\mu_l, \Sigma_l)$. We used a closed-form maximum-likelihood estimation for parameters $\mu_l$, $\Sigma_l$ over source dataset of learned kernels for each layer. We conducted the same experiment as in Section 4.1 for MNIST dataset for this gaussian prior, the results presented at Fig. 5. We found that the gaussian prior performs marginally worse than deep weight prior, log-uniform and standard normal. The gaussian prior could find a bad local optima and fail to approximate potentially multimodal source distribution of learned kernels.

## D  VISUALIZATION OF DEEP WEIGHT PRIOR LATENT SPACE

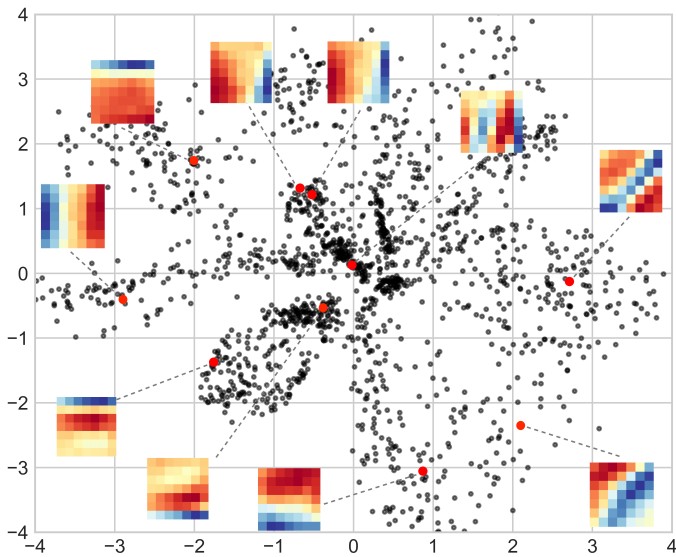

Figure 6: An illustration for Section 4.1. We visualize latent representations of convolutional filters for ConvNet on NotMNIST. Every point corresponds to mean of latent representation $q(z \,|\, w_i)$, where $w_i$ is a kernel of shape $7 \times 7$ from the first convolutional layer, and $q(z \,|\, w_i)$ is an inference network with a two-dimensional latent represenation.

# E  SAMPLES FORM VARIATIONAL AUTO-ENCODERS

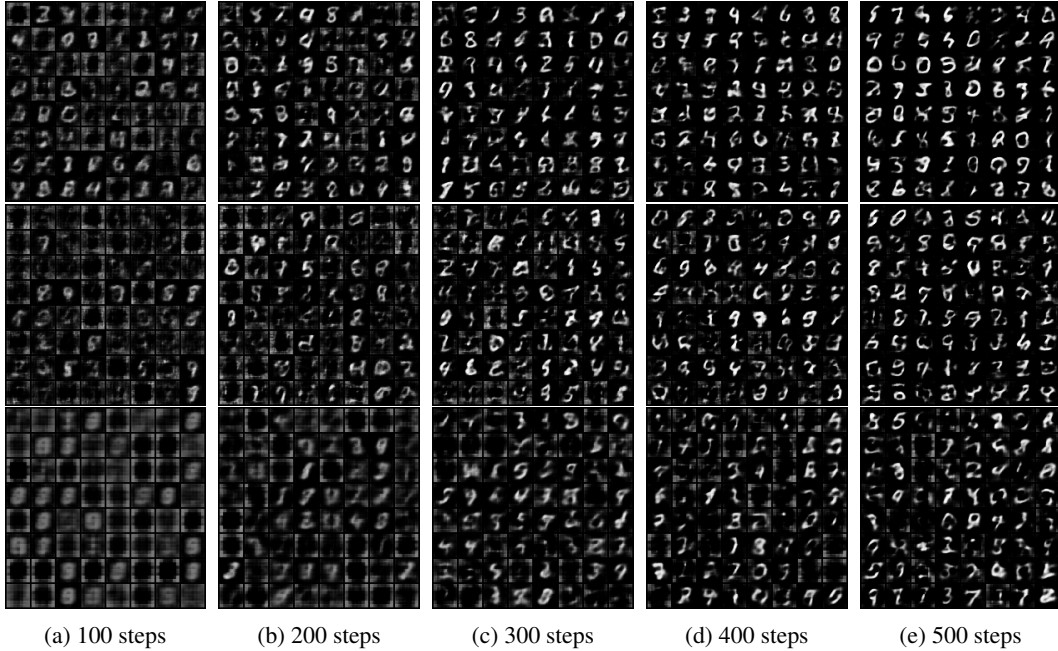

|         (a) 100 steps         |         (b) 200 steps         |         (c) 300 steps         |         (d) 400 steps         |         (e) 500 steps         |

Figure 7: An illustration for the Section 4.3 of samples from variational auto-encoder for three different types of initialization of convolutional layers after 100, 200, 300, 400 and 500 steps of optimization. The first row corresponds to deep weight prior initialization, the second to initialization with learned kernels, and the third to Xavier initialization.

# F  PRIOR ARCHITECTURES

| Encoder5x5 | Decoder5x5 | Encoder7x7 | Decoder7x7 |
|---|---|---|---|
| Conv,  64, $3 \times 3$ | Conv, 128, $1 \times 1$ | Conv, 32, $3 \times 3$ | ConvT, 64, $3 \times 3$ |
| Conv,  64, $3 \times 3$ | ConvT, 128, $3 \times 3$ | Conv, 64, $3 \times 3$ | ConvT, 64, $3 \times 3$ |
| Conv, 128, $3 \times 3$ | ConvT, 128, $3 \times 3$ | Conv, 64, $3 \times 3$ | ConvT, 32, $3 \times 3$ |
| Conv, 128, $3 \times 3$ | ConvT,  64, $1 \times 1$ | $2 \times$ Linear, $z_{dim}$ | $2 \times$ ConvT,  1, $1 \times 1$ |
| $2 \times$ Linear, $z_{dim}$ | $2 \times$ Conv,    1, $1 \times 1$ | | |
| = 260040 params | = 304194 params | = 56004 params | = 56674 params |

Table 2: Architectures of variational auto-encoders for prior distributions. On the left for filters of shapes $5 \times 5$ and for filters of shape $7 \times 7$ on the right. See more details at Section 4 and Appendix H.1. All layers were divided with ELU non-literary.

# G  NETWORK ARCHITECTURES

| Classification MNIST | Classification CIFAR | Variational Auto-encoder MNIST |
|---|---|---|
| Conv, 32, $7 \times 7$ | Conv2d, 128, $7 \times 7$ | Conv2d, 64, stride 2, $7 \times 7$ |
| Conv, 128, $5 \times 5$ | Conv2d, 256, $5 \times 5$ | Conv2d, 128, $5 \times 5$ |
| Linear, 10 | Conv2d, 256, $5 \times 5$ | $2\times$ Linear, $z_{hid}$ |
| | Linear, 512 | ConvT, 128, $5 \times 5$ |
| | Linear, 10 | ConvT, 64, stride 2, $5 \times 5$ |
| | | ConvT, 1, stride 2, $5 \times 5$ |
| = 115658 params | = 5759498 params | = 1641665 params |

Table 3: Network Architectures for MNIST and CIFAR-10/CIFAR-100 datasets (Section 4).

## H PYTORCH ARCHITECTURES

### H.1 VAE PRIORS

- VAE model for 7x7 kernels ($z_{dim} = 2$, 300 epochs, Adam optimizer with linear learning rate decay from 1e-3 to 0.):

```
VAE(
  (encoder): Encoder7x7(
    (features): Sequential(
      (0): Conv2d(1, 32, kernel_size=(3, 3), stride=(1, 1))
      (1): ELU(alpha=1.0)
      (2): Conv2d(32, 64, kernel_size=(3, 3), stride=(1, 1))
      (3): ELU(alpha=1.0)
      (4): Conv2d(64, 64, kernel_size=(3, 3), stride=(1, 1))
      (5): ELU(alpha=1.0))
    (fc_mu): Conv2d(64, 2, kernel_size=(1, 1), stride=(1, 1))
    (fc_var): Conv2d(64, 2, kernel_size=(1, 1), stride=(1, 1)))
  (decoder): Decoder7x7(
    (decoder): Sequential(
      (0): ConvTranspose2d(2, 64,
      kernel_size=(3, 3), stride=(1, 1))
      (1): ELU(alpha=1.0)
      (2): ConvTranspose2d(64, 64,
      kernel_size=(3, 3), stride=(1, 1))
      (3): ELU(alpha=1.0)
      (4): ConvTranspose2d(64, 32,
      kernel_size=(3, 3), stride=(1, 1))
      (5): ELU(alpha=1.0))
    (fc_mu): Conv2d(32, 1, kernel_size=(1, 1), stride=(1, 1))
    (fc_var): Conv2d(32, 1, kernel_size=(1, 1), stride=(1, 1))))
```

- VAE model for 5x5 kernels ($z_{dim} = 4$, 300 epochs, Adam optimizer with linear learning rate decay from 1e-3 to 0.):

```
VAE(
  (encoder): Encoder5x5(
    (features): Sequential(
      (0): Conv2d(1, 64, kernel_size=(3, 3),
      stride=(1, 1), padding=(1, 1))
      (1): ELU(alpha=1.0)
      (2): Conv2d(64, 64, kernel_size=(3, 3),
      stride=(1, 1), padding=(1, 1))
      (3): ELU(alpha=1.0)
      (4): Conv2d(64, 128, kernel_size=(3, 3),
      stride=(1, 1))
      (5): ELU(alpha=1.0)
      (6): Conv2d(128, 128, kernel_size=(3, 3),
      stride=(1, 1))
      (7): ELU(alpha=1.0))
    (fc_mu): Conv2d(128, 4, kernel_size=(1, 1), stride=(1, 1))
    (fc_sigma): Conv2d(128, 4, kernel_size=(1, 1), stride=(1, 1)))
  (decoder): Decoder5x5(
    (activation): ELU(alpha=1.0)
    (decoder): Sequential(
      (0): Conv2d(4, 128, kernel_size=(1, 1), stride=(1, 1))
      (1): ELU(alpha=1.0)
      (2): ConvTranspose2d(128, 128,
      kernel_size=(3, 3), stride=(1, 1))
      (3): ELU(alpha=1.0)
      (4): ConvTranspose2d(128, 128,
      kernel_size=(3, 3), stride=(1, 1))
      (5): ELU(alpha=1.0)
      (6): Conv2d(128, 64, kernel_size=(1, 1), stride=(1, 1))
      (7): ELU(alpha=1.0))
    (fc_mu): Sequential(
      (0): Conv2d(64, 1, kernel_size=(1, 1), stride=(1, 1)))
    (fc_var): Sequential(
      (0): Conv2d(64, 1, kernel_size=(1, 1), stride=(1, 1))
    )))
```

## H.2 NOTMNIST-MNIST

- Source models trained on notMNIST (l2=1e-3, 100 epochs, Adam optimizer with linear learning rate decay from 1e-3 to 0.) look as follows:

```
FConvMNIST(
  (features): Sequential(
    (conv1): Conv2d(1, 256, kernel_size=(7, 7), stride=(1, 1))
    (relu1): LeakyReLU(negative_slope=0.01)
    (mp1): MaxPool2d(kernel_size=2,
    stride=2, padding=0, dilation=1, ceil_mode=False)
    (conv2): Conv2d(256, 512, kernel_size=(5, 5), stride=(1, 1))
    (relu2): LeakyReLU(negative_slope=0.01)
    (mp2): MaxPool2d(kernel_size=2,
    stride=2, padding=0, dilation=1, ceil_mode=False)
    (flatten): Flatten())
  (classifier): Linear(in_features=4608,
  out_features=10, bias=True))
```

- The final model (deterministic) trained on MNIST looks as follows (Adam optimizer with linear learning rate decay from 1e-3 to 0.):

```
FConvMNIST(
  (features): Sequential(
    (conv1): Conv2d(1, 32, kernel_size=(7, 7), stride=(1, 1))
    (relu1): LeakyReLU(negative_slope=0.01)
    (mp1): MaxPool2d(kernel_size=2, stride=2,
    padding=0, dilation=1, ceil_mode=False)
    (conv2): Conv2d(32, 128, kernel_size=(5, 5), stride=(1, 1))
    (relu2): LeakyReLU(negative_slope=0.01)
    (mp2): MaxPool2d(kernel_size=2,
    stride=2, padding=0, dilation=1, ceil_mode=False)
    (flatten): Flatten())
  (classifier): Linear(in_features=1152,
  out_features=10, bias=True))
```

- The final model (bayesian) trained on MNIST looks as follows (Adam optimizer with linear learning rate decay from 1e-3 to 0.):

```
FConvMNIST(
  (features): Sequential(
    (conv1): BayesConv2d(
      (mean): Conv2d(1, 32, kernel_size=(7, 7), stride=(1, 1))
      (var): LogScaleConv2d(1, 32,
      kernel_size=(7, 7), stride=(1, 1), bias=False))
    (relu1): LeakyReLU(negative_slope=0.01)
    (mp1): MaxPool2d(kernel_size=2, stride=2,
    padding=0, dilation=1, ceil_mode=False)
    (conv2): BayesConv2d(
      (mean): Conv2d(32, 128, kernel_size=(5, 5), stride=(1, 1))
      (var): LogScaleConv2d(32, 128,
      kernel_size=(5, 5), stride=(1, 1), bias=False))
    (relu2): LeakyReLU(negative_slope=0.01)
    (mp2): MaxPool2d(kernel_size=2,
    stride=2, padding=0, dilation=1, ceil_mode=False)
    (flatten): Flatten())
  (classifier): Linear(in_features=1152,
  out_features=10, bias=True))
```

## H.3 CIFAR

- The source model for CIFAR looks as follows (l2=1e-4, 300 epochs, Adam, Linear learning rate decay from 1e-3 to 0.):

```
CIFARNet(
  (features): Sequential(
    (conv1): Conv2d(3, 128, kernel_size=(7, 7), stride=(1, 1))
    (bn1): BatchNorm2d(128,
        eps=1e-05, momentum=0.1,
        affine=True, track_running_stats=True)
    (relu1): LeakyReLU(negative_slope=0.01)
```

```
        (maxpool): MaxPool2d(
           kernel_size=2, stride=2,
           padding=0, dilation=1, ceil_mode=False)
        (conv2): Conv2d(128, 256, kernel_size=(5, 5), stride=(1, 1))
        (bn2): BatchNorm2d(
           256, eps=1e-05, momentum=0.1,
           affine=True, track_running_stats=True)
        (relu2): LeakyReLU(negative_slope=0.01)
        (conv3): Conv2d(256, 256, kernel_size=(5, 5), stride=(1, 1))
        (bn3): BatchNorm2d(256,
           eps=1e-05, momentum=0.1,
           affine=True, track_running_stats=True)
        (relu3): LeakyReLU(negative_slope=0.01)
        (conv4): Conv2d(256, 512, kernel_size=(5, 5), stride=(1, 1))
        (bn4): BatchNorm2d(512,
           eps=1e-05, momentum=0.1,
           affine=True, track_running_stats=True)
        (relu4): LeakyReLU(negative_slope=0.01)
        (flatten): Flatten())
      (classifier): Sequential(
        (fc1): Linear(in_features=512, out_features=512, bias=True)
        (bn1): BatchNorm1d(512,
           eps=1e-05, momentum=0.1,
           affine=True, track_running_stats=True)
        (relu1): LeakyReLU(negative_slope=0.01)
        (linear): Linear(in_features=512, out_features=100, bias=True)
    ))
```

- The final deterministic model (for CIFAR10) looks as follows:

```
    CIFARNetNew(
      (features): Sequential(
        (conv1): Conv2d(3, 128, kernel_size=(7, 7), stride=(1, 1))
        (relu1): LeakyReLU(negative_slope=0.01)
        (maxpool): MaxPool2d(kernel_size=2, stride=2,
                 padding=0, dilation=1, ceil_mode=False)
        (conv2): Conv2d(128, 256, kernel_size=(5, 5), stride=(1, 1))
        (relu2): LeakyReLU(negative_slope=0.01)
        (conv3): Conv2d(256, 256, kernel_size=(5, 5), stride=(1, 1))
        (relu3): LeakyReLU(negative_slope=0.01)
        (flatten): Flatten())
      (classifier): Sequential(
        (fc1): Linear(in_features=6400, out_features=512, bias=True)
        (relu1): LeakyReLU(negative_slope=0.01)
        (linear): Linear(in_features=512, out_features=10, bias=True)
    ))
```

- The final Bayesian model (for CIFAR10) looks as follows:

```
    CIFARNetNew(
      (features): Sequential(
        (conv1): BayesConv2d(
          (mean): Conv2d(3, 128, kernel_size=(7, 7), stride=(1, 1))
          (var): LogScaleConv2d(3, 128,
          kernel_size=(7, 7), stride=(1, 1), bias=False))
        (relu1): LeakyReLU(negative_slope=0.01)
        (maxpool): MaxPool2d(kernel_size=2, stride=2, padding=0,
        dilation=1, ceil_mode=False)
        (conv2): BayesConv2d(
          (mean): Conv2d(128, 256, kernel_size=(5, 5), stride=(1, 1))
          (var): LogScaleConv2d(128, 256, kernel_size=(5, 5),
          stride=(1, 1), bias=False))
        (relu2): LeakyReLU(negative_slope=0.01)
        (conv3): BayesConv2d(
          (mean): Conv2d(256, 256, kernel_size=(5, 5), stride=(1, 1))
          (var): LogScaleConv2d(256, 256,
          kernel_size=(5, 5), stride=(1, 1), bias=False))
        (relu3): LeakyReLU(negative_slope=0.01)
        (flatten): Flatten())
      (classifier): Sequential(
        (fc1): Linear(in_features=6400, out_features=512, bias=True)
        (relu1): LeakyReLU(negative_slope=0.01)
        (linear): Linear(in_features=512, out_features=10, bias=True)))
```

