# OpenReview forum: "The Deep Weight Prior"
_ICLR.cc/2019/Conference_

### Official Review · AnonReviewer1 · 2018-10-30
**Deep weight prior**

**Rating:** 7
**Confidence:** 3

**Review:**

This paper considers learning informative priors for convolutional neural network models based on fits to data sets from similar problem domains.  For trained networks on related datasets the authors use autoencoders to obtain an expressive prior on the filter weights, with independence assumed between different layers.  The resulting prior is generative and its density has no closed form expression, and a novel variational method for dealing with this is described.  Some empirical comparisons of the deep weight prior with alternative priors is considered, as well as a comparison of deep weight samples for initialization with alternative initialization schemes.

This is an interesting paper.  It is mostly clearly written, but there is a lack of detail in Section 4 that makes it hard for me, at least, to understand exactly what was done there.  I think the originality level of the paper is high.  The issue of informative priors in these complex models seems wide open and the authors provide an interesting approach both conceptually and computationally.  I did wonder whether there was any link between the suggested priors and the idea of modelling the current and related data sets used in constructing the prior jointly, with data set specific parameters given an exchangeable prior?  This would be a standard hierarchical modelling approach.  Such an approach would not be computationally attractive, I just wondered if there is some conceptual link with the current method being an approximation of that approach in some sense.  In Section 4.1, it seems that for the trained networks on the source datasets, point estimates of the filter weights are treated as data for learning the variational autoencoder - is that correct?  Could you model dataset heterogeneity here as well?  Presumably the p_l(z) density is N(0,I)?  Details of the inference and reconstruction networks are sketchy.  In Section 4.2, you say that the number of filters is proportional to the scale parameter k and that you vary k.  What scale parameter do you mean?

---

> ### Author Response · Authors · 2018-11-14
> **Response to AnonReviewer1**
>
> We would like to thank you for the thoughtful review with interesting suggestions, we will address your questions and concerns below:
>
> 1) A lack of detail in Section 4
>
> We had significantly improved Section 4. Subsection 4.1 has been rewritten and moved to Section 3, as it has more to do with the method rather than experiments. The description of experiments also has been improved.
>
> 2)  I did wonder whether there was any link between the suggested priors and the idea of modelling the current and related data sets used in constructing the prior jointly, with data set specific parameters given an exchangeable prior? … I just wondered if there is some conceptual link with the current method being an approximation of that approach in some sense.
>
> The suggestion indeed seems to be a generalization of the current approach, that however may be closer to empirical Bayes, since the prior distribution p(W) can be adopted using the current dataset. In the paper, we do not consider any explicit model of a dataset e.g., dataset-embeddings, but it would be an interesting direction for research.
>
> 3) In Section 4.1, it seems that for the trained networks on the source datasets, point estimates of the filter weights are treated as data for learning the variational autoencoder - is that correct?
>
> Yes, this is correct.
>
> 4) Could you model dataset heterogeneity here as well?
>
> If we understand the question correctly, given the current model, the answer is no. Currently, we do not explicitly use kernels heterogeneity e.g., (a) the estimated variance of each training kernel, or (b) the problem from which each training kernel comes from, but using these additional factors could be an interesting topic of investigation.
>
> 3) Presumably the p_l(z) density is N(0,I)?
>
> Yes, this is correct, we described the form of the prior p(z) in VAE section, but it is worth to add it explicitly at the description of our model.
>
> 3) In Section 4.2, you say that the number of filters is proportional to the scale parameter k and that you vary k.  What scale parameter do you mean?
>
> We scale a number of filters on every convolutional layer by k (in our experiments k can be equal to 1/16, 1/8, 1/4, 1/2 or 1). This allows us to track how the size of the model influence our method.

---

### Official Review · AnonReviewer2 · 2018-11-01
**Solid Idea with Focused Experiments**

**Rating:** 8
**Confidence:** 4

**Review:**

Summary:

This paper proposes the ‘deep weight prior’: the idea is to elicit a prior on an auxiliary dataset and then use that prior over the CNN filters to jump start inference for a data set of interest.  Both explicit and implicit priors are considered, with the latter having the benefit of increased flexibility but having the drawback of a lack of a parametric form to plug in to the ELBO.  The authors address this last point by extending the ELBO appropriately.  Experiments are performed testing the prior’s ability to capture trained filters (Figure 1), provide a good initialization (Figure 2), improve sample efficiency (Figure 3), improve training speed (Figure 4).

Pros:

I like this paper: it is a intuitive idea, and the experiments explore exactly what one would hope to gain from the prior (i.e. better initialization, improved sample efficiency).  I find the paper clearly written and to have a logical flow.  Furthermore, I think eliciting priors---while so crucial in more traditional Bayesian modeling---has been mostly overlooked by the Bayesian ML community, and this paper clearly shows that there are gains to be had from a fairly straightforward procedure.


Cons:

The only potential issue with the paper is the use of the implicit prior, as it complicates variational inference, requiring the extension to the ELBO described in Section 3.2.  As far as I can tell, all experiments use the implicit priors.  I would have liked to have seen an experiment using a parametric prior (eg Gaussian) that shows what gains the implicit prior provides.  Or is it simply a matter of memory efficiency?


Other comments:

-- Nice first sentence in the introduction!  I like how it’s a general statement but immediately focuses the reader’s attention to the paper’s topic.

-- While it doesn’t say so explicitly, the paper seems to imply it is the first to use implicit priors.  Some previous work that uses some form of implicit prior includes:

Runs a chain to refine the prior: Alex Lamb et al. "GibbsNet: Iterative Adversarial Inference for Deep Graphical Models." Advances in Neural Information Processing Systems. 2017.

Optimizes a NN implicit prior based on an invariance objective: Eric Nalisnick and Padhraic Smyth. "Learning priors for invariance." International Conference on Artificial Intelligence and Statistics. 2018.

Defines implicit priors over functions through samplers: Chao Ma, Yingzhen Li, and José Miguel Hernández-Lobato. "Variational Implicit Processes." arXiv preprint arXiv:1806.02390 (2018).


Evaluation:  I recommend this paper for acceptance.  It is a sensible idea with pointed experimental validation.

---

> ### Author Response · Authors · 2018-11-14
> **Response to AnonReviewer2**
>
> We would like to thank you for the thoughtful review and useful points related to parametric priors and literature on implicit priors, we will address your questions below:
>
> 1) I would have liked to have seen an experiment using a parametric prior (eg Gaussian) that shows what gains the implicit prior provides. Or is it simply a matter of memory efficiency?
>
> We had also evaluated a multivariate Gaussian prior, that shares parameters across all kernels within the specific layer (Gaussian prior over kernels). This prior, however, leads to marginally worse performance in both the likelihood of the generative model and the final accuracy comparing to the VAE-based approximation. We will append these results in the Appendix.
>
> 2) While it doesn’t say so explicitly, the paper seems to imply it is the first to use implicit priors.  Some previous work that uses some form of implicit prior includes
>
> You are right, a number of works have already used implicit priors. We clarified explicitly that the paper is not proposing to use implicit priors but propose a new inference technique that is compatible with Stochastic Gradient Variational Lower Bound. These papers are indeed relevant and we will definitely cite them.

---

> > ### Comment · AnonReviewer2 · 2018-11-26
> > **Response to Response**
> >
> > Thanks for addressing my comments!  My score of 8 remains unchanged.  Good luck with the submission.

---

### Official Review · AnonReviewer3 · 2018-11-02
**Not a strong paper**

**Rating:** 6
**Confidence:** 4

**Review:**

This paper considers modeling convolutional neural network by a Bayes method. The prior for the weights is considered in which the weights from various layers, input and output channels are assumed to be independent.  A varational method is considered to approximate the posterior distribution of the weights of CNN.  It looks to me that the prior distribution is a fairly standard product which may not perfectly suitable for CNN. Also the validity of the proposed variational method needs further evaluation. Below I summarize my concerns more technically.

1. CNN essentially has a tree structure, i.e., each layer can be viewed as the parent of the next layer. So the consecutive layers should have a sort of dependence. Also, the weights based on the same input channel should all inherit features of that channel. Based on these considerations, is it really reasonable to assume that the random weights are independent?
I agree that independence assumption makes the model and computation easier, but the prior itself should reflect the possible dependence structure of the channels.

2. The KL divergence might not be tractable and so the proposed variational method replaces it with an upper bound. This method highly depends on the assumption that the upper bound of the KL divergence is accurate. Otherwise it is hard to tell that the method really approximates the authentic variational method very well. It would be great if the accuracy of the upper bound can be further evaluated (theoretically and numerically).

---

> ### Author Response · Authors · 2018-11-14
> **Response to AnonReviewer3**
>
> We would like to thank you for the thoughtful review and questions about prior factorization and tightness of L^{aux}, we will address raised concerns below:
>
> 1) Is it really reasonable to assume that the random weights are independent?
>
> Before our work, priors for convolutional layers in Bayesian Deep Learning had a fully-factorized structure, namely, all weights were treated independently. We use more general factorization, now at least there are spatial correlations between the weights in a kernel, but not yet between kernels and between layers. The method allows us to use more flexible priors and can be potentially generalized to more complex dependencies. We discuss the factorization in (Section 6) and consider this direction for the future work.
>
> 2) This method highly depends on the assumption that the upper bound of the KL divergence is accurate.
>
> Indeed, our method depends on the accuracy of the upper bound on KL-divergence or equivalently a gap between an intractable variational lower bound L and a proposed variational lower bound L^{aux}. As we show in (Appendix A, eq. 13), maximization of the L^{aux} minimizes the gap L - L^{aux} = E_q(w) [KL(r(z | w) || p(z | w))] by adjusting parameters of reversed model r(z | w), in other words, the gap reduces as reversed model r(z | w) gets more accurate. The bound is tight if and only if the auxiliary distribution coincides with the exact posterior distribution of the implicit weight model, i.e., r(z | w) = p(z | w). A huge piece of literature on ELBOs relies on this type of argument.
>
> At the same time, there already exists a number of works on applications and analysis of tightness of variational bounds (for instance [1, 2]). Therefore, we did not address the underlined issue directly. One of the most straightforward approaches to providing more accurate variational bounds is to employ importance weighted bounds introduced in [3], which allow to trade off sample complexity and tightness of variational bounds. We have added the estimates of approximation gap using importance weighted bound to Appendix F, however, these figures should be treated with caution, as IWAE still gives us a lower bound on L.
>
> [1] On the Quantitative Analysis of Decoder-Based Generative Models, https://arxiv.org/abs/1611.04273
> [2] Tighter Variational Bounds are Not Necessarily Better, https://arxiv.org/abs/1802.04537
> [3] Importance Weighted Autoencoders, https://arxiv.org/abs/1509.00519

---

> > ### Comment · AnonReviewer3 · 2018-11-26
> > **response to rebuttal**
> >
> > Thanks for your response. I still believe that a prior demonstrating dependence across layers and nodes is very important as this is the nature of Bayesian deep network. Also it is important to comment on some relating results on the upper bound in your approximation.

---

> > > ### Comment · AnonReviewer2 · 2018-11-27
> > > **Dependence Structure of Prior**
> > >
> > > I think that Reviewer #3 is being too critical of the prior's lack of structure, and I don't agree with the point that "this is the nature of Bayesian deep network[s]."  The automatic relevance determination (ARD) prior [1] is the only structured one that I know of for Bayesian NNs, and its use is somewhat rare since performing inference for the scale hyperprior is challenging.  Just about all papers on Bayesian NNs use fully factorized priors, and specifying inter-layer priors is a completely open problem, as far as I'm aware.
> > >
> > > [1] MacKay, David JC. "Bayesian non-linear modeling for the prediction competition." Maximum Entropy and Bayesian Methods.  1996. 221-234.

---

> > > > ### Comment · AnonReviewer3 · 2018-11-27
> > > > **More references on structured prior in Bayesian NNs**
> > > >
> > > > The following paper introduces another prior, which allows dependence across the neurons.
> > > >
> > > > [1]  Shengyang Sun, Changyou Chen, Lawrence Carin. "Learning Structured Weight Uncertainty in Bayesian Neural Networks."  Proceedings of the 20th International Conference on Artificial Intelligence and Statistics, PMLR 54:1283-1292, 2017.

---

> > > > > ### Comment · AnonReviewer2 · 2018-11-27
> > > > > **Re: More References**
> > > > >
> > > > > Sure, intra-layer dependencies can be done---as the deep weight prior does (well, intra-kernel dependencies in this case).  Extending these priors across layers is the hard part, which this reference still doesn't do.

---

### Meta-Review · Area_Chair1 · 2018-12-16
**Good paper, but need to some discussion**

**Confidence:** 5
**Recommendation:** Accept (Poster)

**Metareview:**

This paper proposes factorized prior distributions for CNN weights by using explicit and implicit parameterization for the prior. The paper suggest a few tractable methods to learn the prior and the model jointly. The paper, overall, is interesting.

The reviewers have had some disagreement regarding the effectiveness of the method. The factorized prior may not be the most informative prior and using extra machinery to estimate it might deteriorates the performance. On the other hand, estimating a more informative prior might be difficult. It is extremely important to discuss this trade-off in the paper. I strongly recommend for the authors to discuss the pros and cons of using priors that are weakly informative vs strongly informative.

The idea of using a hierarchical model has been around, e.g., see the paper on "Hierarchical variational models" and more recently "semi-implicit Variational Inference". Please include a related work on such existing work. Please discuss why your proposed method is better than these existing methods.

Conditioned on the two discussions added to the paper, we can accept it.